# Microbiota Transplantation in an Antibiotic-Induced Bacterial Depletion Mouse Model: Reproducible Establishment, Analysis, and Application

**DOI:** 10.3390/microorganisms10050902

**Published:** 2022-04-26

**Authors:** Lijun Shang, Jiayu Tu, Ziqi Dai, Xiangfang Zeng, Shiyan Qiao

**Affiliations:** 1State Key Laboratory of Animal Nutrition, Ministry of Agriculture Feed Industry Centre, China Agricultural University, Beijing 100193, China; shanglj1996@163.com (L.S.); s20213040636@cau.edu.cn (J.T.); daiziqi@cau.edu.cn (Z.D.); ziyangzxf@163.com (X.Z.); 2Beijing Bio-Feed Additives Key Laboratory, Beijing 100193, China

**Keywords:** antibiotic, application, FMT, microbiota, protocol

## Abstract

The fecal bacteria transplantation (FMT) technique is indispensable when exploring the pathogenesis and potential treatments for microbiota-related diseases. For FMT clinical treatments, there are already systematic guidelines for donor selection, fecal bacterial separation, FMT frequency, and infusion methods. However, only a few studies have demonstrated the use of standardized FMT procedures for animal models used in theoretical research, creating difficulties for many new researchers in this field. In the present paper, we provide a brief overview of FMT and discuss its contribution to the current understanding of disease mechanisms that relate to microbiota. This protocol can be used to generate a commonly used FMT mouse model and provides a literature reference of customizable steps.

## 1. Introduction

The animal body is inherently metagenomic, not only in relation to the eukaryotic genome that makes up the body, but also the genomes of the microbiomes colonizing the surface of the body, which includes bacteria, archaea, fungi, protozoa, and viruses [1]. A growing body of research has shown that commensal microbial communities interact with almost all physiological aspects of the host in health and disease [2,3,4,5,6,7]. The microbiome within the gut is the most widely studied because its microbial biomass exceeds that of other bodily habitats by a large order of magnitude, and it is separated from the host only by a single layer of epithelial cells [5]. Numerous studies have shown that gut microbiota dysbiosis contributes to the onset of many diseases, from gastrointestinal and metabolic disorders to immune and neuropsychiatric diseases [8,9,10,11].

In this context, fecal microbiota transplantation (FMT), an important means of regulating the composition and functions of the gut microbiota [12], is often used in studies of the gut microbiota. Clinically, FMT, also known as fecal transplantation, is a procedure in which stool from a healthy donor is placed into another patient’s intestine [13]. However, in experimental studies, its definition is broader. Common research modes include transplanting wild-type (WT) mice or healthy human microbiota into disease-model recipients, [14,15,16] transplanting disease model microbiota into recipients [17,18,19], or even transplanting a customized combination of microbiota (selective microbiota transplantation, SMT) to achieve specific experimental purposes [20]. Some studies have also applied a combination of several modes, for example, the inclusion of a model group that acts as both the donor and recipient to control for handling and allows for the analysis of confounding factors that may affect the experimental groups [21].

In clinical situations, there are already systematic FMT treatment guidelines in place that are constantly being updated and improved [22,23,24]. There have been many reports on donor selection, the separation of fecal bacteria, the frequency of FMT, and infusion procedures [13,22]. Nevertheless, only a few studies have been designed to explore the methodology of FMT, and even fewer have provided standardized FMT procedures for use in animal models used in research [25], creating difficulties for many beginners in microbiota research. Therefore, the aim of the present review is to provide a simple and repeatable FMT protocol, as well as a summary of the literature references for each adaptable step to aid in customizing microbiota. Moreover, necessary analyses related to FMT, as well as common patterns among studies that have developed this technique to investigate the disease mechanisms related to the intestinal ecosystem, are discussed.

## 2. The Development and Overview of the Procedure

The history of using stool from healthy people to treat human diseases dates back to the fourth century AD [26]. Hong Ge, a Chinese doctor during the Dong Jin Dynasty (AD 300–400), recorded the treatment of “Wen Bing” (febrile disease) and “Shang Han” (typhoid fever) by drinking the fecal suspension or fermented feces [27,28]. Later, in the Compendium of Materia Medica, which is the most comprehensive record of resolving diseases in traditional medicine, Shizhen Li described more than 20 indications that can be effectively treated with fecal suspension or fermented feces [29]. In 1958, Eiseman et al. successfully treated patients with severe pseudomembranous enteritis using a fecal suspension, which was the first recorded instance of such a treatment in the English literature [30]. In 2011, the method was officially termed fecal microbiota transplantation (FMT) [31], and, in 2013, it was included in the medical guidelines for the treatment of refractory *Clostridiodies difficile* infection (CDI), which represented a milestone in the history of FMT application [32]. The most successful application of FMT, to date, has been in the treatment of refractory CDI [22,33], and there is growing evidence that FMT also has great potential for interventions in other enteric-related diseases and in neurological conditions [14,20,34,35,36].

FMT, as a strategy to modulate gut microbes, is not only a breakthrough medical technique, but also a breakthrough in technological and theoretical research. In theoretical studies, interest in FMT has not been limited to its use as a therapeutic method (in which the fecal microflora from healthy donors is transmitted to patients to restore a healthy microbial composition to the gut), because it has also involved the transfer of bacteria from one individual to another to induce a desired physiologic effect. Potential microbial material from donors is not limited to feces, and may also comprise intestinal contents or specially modified microbiota (such as SMT) [19,20]. Researchers have often applied FMT to studies of mouse models, which, because of their genetic proximity to humans, our ability to genetically manipulate their genomes, and the availability of many tools, mutants, and inbred strains, have become the mammalian model of choice [37,38]. Therefore, recipients are not limited to germ-free mice and may also be genetically engineered mouse models [15,39].

Of all the available research methods, the most basic is the transplantation of the microbiota of target mouse donors into antibiotic-induced bacterial-depletion mouse recipients, and other research models can be modified on this basis. In Section 3 and Section 4 of this review, we provide a brief overview of the selected models of FMT (mouse donors; antibiotic-induced bacterial-depletion mouse recipients), along with the literature references and considerations for each step. In the following sections, we provide some of the necessary analyses related to FMT. Finally, we discuss the contribution that FMT has made to the current understanding of disease mechanisms related to gut microbiota.

## 3. Protocol

A schematic of all the steps can also be observed in Figure 1.

### 3.1. Materials

(1)Phosphate-buffered saline (PBS), sterile.(2)Sample storage buffer: Glycerin in PBS = 1:1 (*v*/*v*), sterilization (see Note 1).(3)Placebo: Glycerin in PBS = 1:4 (*v*/*v*), sterilization (see Note 2).(4)Broad-spectrum antibiotic mix: 500 mg/L of vancomycin, 1 g/L of ampicillin, and 1 g/L of neomycin (see Note 3).(5)A 40 μm cell strainer.(6)Lavage needle, size 9 (Table 1) (see Note 4).(7)A total of 2 mL of cryotubes, round bottom (see Note 5).(8)A 1 mL syringe (see Note 6).(9)Cage, tweezers, cryotubes, and all other appliances and reagents need to be sterilized before use.

### 3.2. Procedure

#### 3.2.1. Donors

(1)All the donors are raised in separate cages. Place all the regents and buffers on ice.(2)Weigh the dry, empty 2 mL cryotubes.(3)Collect fresh feces in cryotubes and weigh (see **Note 7**).(4)Add an appropriate amount of pre-cooled sterile PBS (V_1_) to produce a feces concentration of 50–100 mg/mL. Reach as close to the upper limit as possible.(5)Homogenize the solution at 4 °C.(6)Filter the solution through a 40 μm filter and collect the supernatant in a new tube.(7)Centrifuge at 8000× *g* for 5 min at 4 °C.(8)Discard the supernatant without disturbing the sediment.(9)Transplantation solution: Resuspend the pellet obtained from the steps above in cold PBS (the original volume, V_1_), and add an equal volume of sample storage buffer (see **Note 1**).(10)If not used immediately, freeze in liquid nitrogen and store at −80 °C.

#### 3.2.2. Recipients

(1)Administer to the recipients the broad-spectrum antibiotic mix instead of drinking water for at least 14 days (see **Note 8**), and allow to “rest” for 1–2 days (see **Note 9**).(2)Before intragastric administration, fast all recipients but allow them to drink freely for 1 day (see **Note 10**).(3)Intragastrically administer each recipient with 200–300 μL (see **Note 6**) of transplantation solution once a day for 5 consecutive days (Table 2) (see **Note 11**).

## 4. Notes

(1)When the concentration of glycerol in the sample storage buffer is 50%, add the sample storage buffer to the resuspended solution at a ratio of 1:1 to prepare the transplantation solution and obtain a final glycerol concentration of 25%. The final concentration glycerol can be adjusted within the range of 10–30% according to different experimental purposes and conditions.(2)Use a placebo as a control for the transplantation solution. The final glycerol concentration in the placebo should be equal to that of the transplantation solution.(3)Metronidazole can be added to the broad-spectrum antibiotic mix, but it must be used with caution. This can achieve better bacterial depletion results, but may cause weight loss in mice [42,43]. You can try to gradually introduce metronidazole to the solution [42].(4)The procedure is suitable for commonly used 6-to-8-week-old mice (25–30 g). If using other target animals, refer to Table 1 and Table 2.(5)Compared to a tapered-bottom tube, using a round-bottom tube can achieve better homogenization effects.(6)A 1 mL syringe is suitable for commonly used 6-to-8-week-old mice (25–30 g). If using other target animals, refer to Table 2.The volume received by each recipient (200–300 μL) is calculated for approximately 10–20 mg feces/mouse. Generally, the ideal effect can be obtained from a feces concentration of 50–100 mg/mL in 200 μL of transplantation solution. To determine the optimum conditions for a particular model, a pilot experiment is required.(7)Generally, each mouse can provide 50–100 mg of fresh feces (6-to-8-week-old mice, 25–30 g). However, if the donors are enteritis-mouse models, there will be less feces. Therefore, the stool from mice of matched weight and sex can be mixed depending on the experimental design. The number of animals used can be customized to the experiment.Fresh feces should be used for transplantation within 6 h [22,36,44], as oxygen exposure degrades the fecal bacterial communities [45]. If frozen feces are required for subsequent use, aim to complete the freezing operation within 15 min as much as possible [46].(8)The duration of the broad-spectrum antibiotic mix treatment of mice can be customized, but, generally, it lasts for at least 14 days.(9)Before FMT, a “rest” period of 12–48 h is required [22,36,42,44].(10)Fasting should be started for at least 4–8 h before gavage to avoid the gastric contents hindering gavage injection and affecting the drug absorption rate.(11)The transplanting of recipients once a day for 5 consecutive days is suitable for commonly used 6-to-8-week-old mice (25–30 g). To determine the optimum conditions for a particular model, a pilot experiment is required.For the solutions to other common problems, see Table 3.

## 5. Microbiological Analyses

### 5.1. The Detection of Donors

The detection of the donor microbiota is a necessary stage. First, the homogeneity of the donor microbiota can be determined to avoid a great degree of variability within recipient groups after FMT. Second, the composition of the donor microbiota is used to verify the successful separation of fecal bacteria.

### 5.2. The Detection of the Transplantation Solution

First, the similarity between the microbiota of transplantation solution and that of the donor is detected, because a low similarity indicates that the separation of the flora has failed. Second, the microbial composition of the transplantation solution is clarified, which provides a data reference for subsequent analysis and experiments.

Most studies suggest that the microbiota in recipients after FMT tends towards the donor composition [35,47,48,49]. Therefore, this test can also evaluate the success of FMT.

### 5.3. The Detection of the Recipients

#### 5.3.1. Baseline

Before fecal transplantation, baseline microbial composition testing is critical. Studies have shown that the recipient’s microbial diversity at baseline predicts their responses to FMT [50]. Generally, broad-spectrum antibiotic mix treatment depletes mice of their intestinal microbiota by hundreds of times to a level similar to that observed in germ-free mice [42,43].

#### 5.3.2. After FMT

After FMT, 16S rRNA gene sequencing is recommended for both the FMT and placebo group to identify the microbiota. In general, a statistically significant change in the microbiota composition in the FMT group, compared to the placebo group, was found after treatment [34].

Most studies compare the changes in the flora of fecal samples to determine the success of FMT [44]. Ishikawa et al. stated that the feces, luminal contents, or mucosa of the target intestinal segment can be analyzed on the basis of the needs of the experiment, and even the sites of the sampling can be chosen as microbial donor sites. Different detection sites illustrate different mechanisms. Stool is almost identical to the contents of the rectum, and the contents of a particular intestinal site are more reflective of the physiological conditions at that particular location in the intestine than feces. Their mucosa provides the best reflection of the colonization of microorganisms and, due to their proximity, may have an advantage in reflecting microbial interactions within the enteric nervous system or mucosal immune system.

## 6. Application

Research into FMT, combined with sequencing, bioinformatics techniques, and an up-to-date holistic understanding of the microbiome, provides new intuitive evidence for the treatment and mechanisms of microbiota-related diseases.

### 6.1. Healthy Individuals as Donors

In general, this method of applying FMT is used to transplant the microbiota of healthy individuals to regulate the microbiota of diseased recipients, thereby verifying the therapeutic effect of the microbiota on the disease. For example, Ishikawa et al. found that FMT, following antibiotic pretreatment with Amoxicillin, Fosfomycin and Metronidazole, may be useful for the treatment of ulcerative colitis [44]. Claudia et al. transplanted microbiota from normal donors into a dextran sodium sulphate-induced colitis mouse model and found that restoring a normobiotic core ecology contributed to the resolution of inflammation [14].

Researchers often combine FMT with other techniques to further elucidate the underlying mechanisms of FMT treatment of diseases. Using the sorting and sequencing of immunoglobulin (Ig) A-coated microbiota (called IgA-seq) techniques, Lima et al. identified immune-reactive microbiota during FMT [49].

In recent years, the active communication between the gut microbiota and the nervous system was discovered [9,10,51]. Studies have shown that FMT treatment can improve abnormal gut microbiota and cognitive deficits and, therefore, its potential as a therapeutic strategy for cognitive dysfunction and Alzheimer’s disease (AD) [15,16].

### 6.2. Disease Models as Donors

As the donor, the bacterial community of the disease model is generally significantly different from that of the healthy control, and is often used to study the disease’s characteristic microbial impact on various aspects of the body’s physiology. Studies have shown that donor mice display disease-related phenotypic alterations that can be transferred from donors to recipients by FMT [19,52,53]. Furthermore, other techniques, such as using genetically engineered mice or META analysis, can be combined to further understand the mechanisms of the microbial influence on disease development, such as core flora [19] or immune regulation [49,52]. Furthermore, human donors can be used to transplant microbiota to mice. After the colonization of germ-free mice with hypertensive patient-derived strains, elevated blood pressure was observed and, although needing to be transferred through the microbiota, illustrating a novel causal role for abnormal gut microbiota in the pathogenesis of hypertension [54].

### 6.3. Customized Microbiota as Donors

Once a substance or gene is known to have a positive effect on a disease, then the substance-modified microbiota from donors can be used in FMT as a transplantation solution. For example, a ketogenic diet (KD) is known to be useful in the treatment of refractory epilepsy, but the mechanisms underlying its neuroprotective effects remain unclear. By transplanting the most abundant microbiota from the KD-diet mice into antibiotic-treated mice, Olson et al. revealed a potential mechanism by which the gut microbiota modulates the host’s metabolism and susceptibility to seizures [20]. Similarly, phlorizin (PHZ), a phytonutrient in apples, can promote good body health. Zhang et al. performed FMT by transplanting the feces of PHZ-fed mice to the high-fat-diet (HFD)-fed mice, confirming that feeding HFD mice the gut contents of the PHZ-modulated mice attenuates HFD-induced metabolic disorders [55]. Similar studies also used genetically engineered mice or known disease-tolerant races as bacteria donors [56,57]. Another method is the application of a mixture of several bacteria from a donor. These can be core microbiota found through meta-analysis in previous studies, and their function can be re-validated by FMT [2,58].

### 6.4. A Combination of the above Donors

It is more common to use combinations of multiple models than the above two models alone. Compared with healthy individuals, using disease models as donors can reproduce a disease phenotype in recipient mice [39,59,60,61]. Sharon et al. conducted further studies, including in vivo metabolome and validation tests, proposing that the gut microbiota regulates behaviors in mice via the production of neuroactive metabolites [61]. Similarly, Kundu et al. used metagenome analysis to select sodium butyrate as a candidate metabolite and, on in vivo re-validation, reproduced the phenotype of FMT [62]. Britton et al. combined the results of 16S rRNA gene sequencing with the detection of homeostatic intestinal T-cell responses to interpret a general mechanism for the microbial contribution to inflammatory bowel disease [63].

## 7. Conclusions and Perspectives

Strategically FMT is the most direct method used to change the composition of gut microbiota. In the present study, we provided a brief overview of the FMT protocol and summarized the research progress of FMT. However, this review has some limitations. One limitation is that we only included oral administration, the most common route. The protocol provided in a previous section may not be generalizable to other routes, such as rectal FMT. Additionally, we only provide the primary means of FMT and solutions to common problems, and some other difficulties and innovations are not included. These limitations may mean that the instructions are only informative for beginners. However, with the rapid progress of gut microbiology, it is hoped that more studies will be conducted in the future to further clarify the application prospects of FMT and seek more comprehensive and optimized FMT strategies.

## Figures and Tables

**Figure 1 microorganisms-10-00902-f001:**
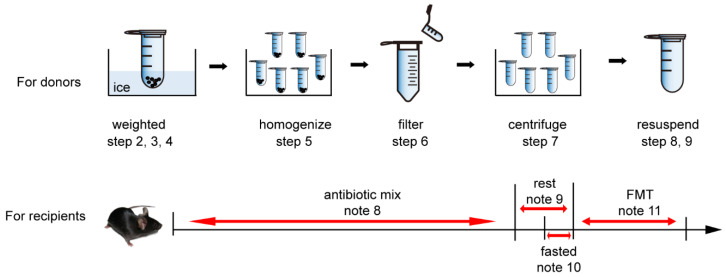
Schematic overview of the models, for which setup is described in the Procedure Section.

**Table 1 microorganisms-10-00902-t001:** Common type and application range of the intragastric needle [40].

Type	Length	Needle Diameter	Apply
6	5 cm	0.6 mm	Nude mice, about 5 weeks old
7	5 cm	0.7 mm	≤25 g mice
8	4.5 cm	0.8 mm	≤30 g mice
9	6 cm	0.9 mm	≤30 g mice
12	4 cm	1.2 mm	~50 g mice
12	5 cm	1.2 mm	~100 g mice
12	6 cm	1.2 mm	~150 g mice
16	8 cm	1.6 mm	150~250 g rat
16	9 cm	1.6 mm	~250 g rats
16	11 cm	1.6 mm	~350 g rats
18	10 cm	1.8 mm	≥350 g rats
20	10 cm	2.0 mm	Rats (≥350 g), dogs, rabbits, and birds

**Table 2 microorganisms-10-00902-t002:** Maximum intragastric intake of commonly used laboratory animals at one time [41].

Animal Species	Weight (g)	Gastric Volume (mL)
Mice	20–24	0.8
	25–30	0.9
	≥30	1.0
Rats	100–199	3.0
	200–249	4–5
	250–300	6.0
	≥300	8.0
Guinea pigs	250–300	4–5
	≥300	6.0
Rabbits	2000–2400	100
	2500–3500	150
	≥3500	200
Cats	2500–3000	50–80
	≥3000	100–150
Dogs	10,000–15,000	200–500
Pigs	-	500

**Table 3 microorganisms-10-00902-t003:** Troubleshooting table.

Problem	Possible Reason	Solution
High incidence of animal death	Unskilled intragastric operation, resulting in excessive stress	Gavage should be painless. If the animal persistently struggles, has difficulty breathing, or resists needle insertion, immediately stop needle insertion and pull the needle out. Try again after the animal has become calm. After the mice have been injected and released, and observations have been conducted for any respiratory abnormalities, the success of the gavage insertion should be confirmed
Low colonization efficiency	(1) Low volume of feces collected; (2) short duration of intragastric administration of bacterial liquid; (3) operation from solution preparation to intragastric administration more than 6 h; (4) and inappropriate glycerin concentration	(1) To increase the amount of feces collected, feces from several mice can be mixed on the basis of experimental needs; (2) intragastric administration should last at least 2 weeks; (3) operation should be fast to reduce the exposure time under oxygen; and (4) the glycerin concentration should be appropriate
High degree of variability within experimental groups	(1) Excessive differences in donor flora; (2) Failed intragastric administration of some recipients	(1) Donors should be rigorously selected, and detection of donor flora is necessary; (2) gavage operations should be skillfully conducted to ensure that mice do not spit out the bacterial fluid

## Data Availability

Not applicable.

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
