# Peer review of "Microbiota Transplantation in an Antibiotic-Induced Bacterial Depletion Mouse Model: Reproducible Establishment, Analysis, and Application"

_microorganisms, 2022, doi:10.3390/microorganisms10050902_

Round 1
Reviewer 1 Report
In this review, Shang et al. provide a step-by-step procedure for fecal microbiota transplant (FMT). The authors identify several areas for trouble shooting and the protocol is very straightforward. It’s not a true review, but it is a helpful protocol for FMT. The manuscript is well written, straight-forward and provides many helpful suggestions. As a result, I have only minor comments:
Comments
- In the materials, it is important to note if the PBS must be sterile
- Please convert rpm to g for universal comparison
- FMT into germ-free mice is also important control of any work with germ-free animals. This point should be added to the application section
Author Response
We would like to thank the reviewers for your professional comments and suggestions, which are very helpful in improving our manuscript. Based on the instructions provided in your letter, we uploaded the file of the revised manuscript. Appended to this letter is our point-by-point response to the comments raised by the reviewers. The comments are reproduced and our responses are given directly afterward in a different color (red). We would like also to thank you for allowing us to resubmit a revised copy of the manuscript.
Response to comment
Reviewer 1
Suggestions 1: In the materials, it is important to note if the PBS must be sterile.
Reply: We feel sorry for the vague statement. Thank you for your nice advice. We have this statement in part 3.1 (9) “Cage, tweezers, cryotubes, and all other appliances and reagents need to be sterilized before use”, but it is not obvious and therefore misleading to the reader. Therefore, according to your suggestion, we added a separate statement that PBS, an important reagent, needs to be sterile (line 88).
Suggestions 2: Please convert rpm to g for universal comparison.
Reply: Thank you so much for your careful check. Follow your suggestion, we revised the manuscript in part 3.2.1 (7), line 108, “Centrifuge at 8000 × g for 5 min”.
Suggestions 3: FMT into germ-free mice is also important control of any work with germ-free animals. This point should be added to the application section.
Reply: Thank you for your rigorous comment. In the application section, we made separate statements based on the type of donor selected in the study. Antibiotic-induced bacterial depletion mouse model or germ-free mice used as donors, the only difference is whether the antibiotic cocktail needs to be treated before the FMT, and it has little effect on the overall experimental design and the application review of the FMT technology. But it must be shown that we also agree that germ-free mice are more capable of exhibiting the effect of the donor microbiota on the body, whereas antibiotic clearance only mimics this pattern, and even mild antibiotic treatment can serve as a model of bacterial disturbance.

Reviewer 2 Report
The aim of the paper was to provide a brief overview of fecal bacteria transplantation technique and discuss its contribution to the current understanding of diseases related to microbiota. According to the authors, the presented protocol can be used to generate a commonly used FMT mouse model. They also provided a literature reference of customizable steps. In my opinion, this comprehensive review is quite interesting for the readers.
My only concerns are:
- Keywords and abbreviations should be listed in alphabetical order.
- An additional section/chapter describing disadvantages/threats of the proposed protocols would be useful if included into the paper.
- Line 41-43, the sentence is confusing, should be rewritten.
- Clostridium difficile is the former name of the bacteria species, the current one should be used.
- Section 3 should be re-arranged somehow, in my opinion. It should not list Materials at the beginning with so many references to the further pieces. It is not very convenient to follow the text arranged like that.
- Line 154 – this is the reference to Table 2, I guess?
- Line 182 – different size of the fonts.
- Reference list - no 24 and no 31 need corrections.
However, all the points mentioned above do not decrease the overall value of the paper.
Author Response
We would like to thank the reviewers for your professional comments and suggestions, which are very helpful in improving our manuscript. Based on the instructions provided in your letter, we uploaded the file of the revised manuscript. Appended to this letter is our point-by-point response to the comments raised by the reviewers. The comments are reproduced and our responses are given directly afterward in a different color (red). We would like also to thank you for allowing us to resubmit a revised copy of the manuscript.
Response to comment
Reviewer 2
Suggestions 1: Keywords and abbreviations should be listed in alphabetical order.
Suggestions 2: An additional section/chapter describing disadvantages/threats of the proposed protocols would be useful if included into the paper.
Suggestions 3: Line 41-43, the sentence is confusing, should be rewritten.
Reply: Thank you for your rigorous comment. Follow your suggestion, we revised keywords and abbreviations parts, and (1) in line 248-251, we have added the following description: This review had some limitations. We have only provided the primary means of FMT and solutions to common problems, and some other difficulties and innovations are not included. These limitations may mean the instructions are only informative for beginners. However, it is hoped that this resource can inspire researchers and lead to more vigorous research into FMT. (2) in line 35-37, we rewritten the “For example, the additional inclusion of a model that acts as both the donor and recipient to control for handling or allow for inter-element comparisons” to “for example, the inclusion of a model group that acts as both the donor and recipient to control for handling and allow for the analysis of confounding factors that may affect the experimental groups”.
Suggestions 4: Clostridium difficile is the former name of the bacteria species, the current one should be used.
Reply: Thank you so much for your rigorous suggestions. Do you mean it should be changed to Clostridioides difficile? For consistency, we simply use the names in the cited literature and ignore this problem. We agree with the suggestion that the latest name should be used in the literature.
Suggestions 5: Section 3 should be re-arranged somehow, in my opinion. It should not list Materials at the beginning with so many references to the further pieces. It is not very convenient to follow the text arranged like that.
Reply: Thank you so much for your constructive suggestions. This paper is written in accordance with the way of preparing for the test, which may have certain limitations. I apologize for the unreadability of the article. This suggestion will be given priority in the subsequent article writing.
Suggestions 6: Line 154 – this is the reference to Table 2, I guess?
Suggestions 7: Line 182 – different size of the fonts.
Suggestions 8: Reference list - no 24 and no 31 need corrections.
Reply: We gratefully appreciate for your nice comment. Based on your suggestions, we have revised the above sections.

Reviewer 3 Report
Fecal microbiota transplantation topic is of interest, and many attempts are ongoing to standardize the procedure of FMT in human practice. This is clinically very relevant.
Authors proposed in this Review to develop a 'standardized model of FMT for experimental approaches' and to do 'summary of literature references'. However, I am overall disturbed by the organization of this Review. It sounds like a protocol paper, with the addition of personal notes. To me, this is not what I am expecting to read for a Review paper in this field. Literature review is up to date, and I see no obvious signs of biased citations.
Specifics:
- As written in the intro, strategies using microbiota transplantation in the laboratory is specially designed to serve specific experimental purposes. I don't necessarily see an interest to provide, 'a simple and repeatable FMT protocol'. For instance, Chapter 4 with authors' notes clearly proves that each parameter is supposed to vary according to scientific purpose, making the whole concept of a generic FMT in laboratory experiments useless...
- To me the most important aspect is to preserve at best the anaerobic environment of samples between donor's feces and recipient. This notion is lacking and it is major.
- Another important missing point of discussion here is the administration route for FMT, what about rectal FMT? Please discuss. This is important.
- Authors' notes and troubleshooting Table 3 is a nice attempt, but I feel overall most points discussed, and their solutions, are very trivial and not much innovative. For instance, when low colonization efficiency, the Reason is 'low volume' and the Solution is 'increase volume' or the reason is 'inappropriate glycerin concentration' the solution is 'use appropriate concentrations'...
I have many comments on the suggested 'standard protocol' discussed here:
Steps 4 and 5: I don't recognize the interest to perform initial steps at 4 degrees.
Steps 8, 9, 10 : What is the interest to discard 40µm-filtered material ? Why snap freezing the fecal pellet in nitrogen ?
Recipients Step 1: I am missing a broad spectrum anaerobic antibiotic in the mix. Agree with authors' note on metronidazole, but it is important if not using GF mice. I don't see the interest of the 'resting' period. Risk is high that microbiota recover 48H after the end of abx.
Author Response
We would like to thank the reviewers for your professional comments and suggestions, which are very helpful in improving our manuscript. Based on the instructions provided in your letter, we uploaded the file of the revised manuscript. Appended to this letter is our point-by-point response to the comments raised by the reviewers. The comments are reproduced and our responses are given directly afterward in a different color (red). We would like also to thank you for allowing us to resubmit a revised copy of the manuscript.
Response to comment
Reviewer 3
Suggestions 1: As written in the intro, strategies using microbiota transplantation in the laboratory is specially designed to serve specific experimental purposes. I don't necessarily see an interest to provide, 'a simple and repeatable FMT protocol'. For instance, Chapter 4 with authors' notes clearly proves that each parameter is supposed to vary according to scientific purpose, making the whole concept of a generic FMT in laboratory experiments useless…
Suggestions 3: Authors' notes and troubleshooting Table 3 is a nice attempt, but I feel overall most points discussed, and their solutions, are very trivial and not much innovative. For instance, when low colonization efficiency, the Reason is 'low volume' and the Solution is 'increase volume' or the reason is 'inappropriate glycerin concentration' the solution is 'use appropriate concentrations'
Reply: We gratefully thanks for the precious time the reviewer spent making constructive remarks. Different experimental designs and selection of experimental animals make it impossible for the conditions of the same experimental method to be consistent. We have clearly given the scope of application, the dosage to refer to when the animal weight/species changes. We do not believe that it is scientifically feasible to provide a universal protocol for all research. Of course, for a microbiome expert like you, the preliminary research protocol does not provide any inspiration. Follow your suggestions, in line 248-251, we also added the following description: This review had some limitations. We have only provided the primary means of FMT and solutions to common problems, and some other difficulties and innovations are not included. These limitations may mean the instructions are only informative for beginners. However, it is hoped that this resource can inspire researchers and lead to more vigorous research into FMT.
Suggestions 2: To me the most important aspect is to preserve at best the anaerobic environment of samples between donor's feces and recipient. This notion is lacking and it is major.
- Another important missing point of discussion here is the administration route for FMT, what about rectal FMT? Please discuss. This is important.
Reply: We gratefully appreciate for your nice comment. We totally agree with you, and described in line 145-146: “Fresh feces should be used for transplantation within 6 hours, as oxygen exposure degrades the fecal bacterial communities”.However, the protocol does have many limitations. Our methods, drawn from our own previous research, provide only a partial reference to the most commonly used approaches. We will delve deeper into the administration route if it is covered in future research.
Suggestions 4: Steps 4 and 5: I don't recognize the interest to perform initial steps at 4 degrees.
Steps 8, 9, 10 : What is the interest to discard 40µm-filtered material ? Why snap freezing the fecal pellet in nitrogen ?
Recipients Step 1: I am missing a broad spectrum anaerobic antibiotic in the mix. Agree with authors' note on metronidazole, but it is important if not using GF mice. I don't see the interest of the 'resting' period. Risk is high that microbiota recover 48H after the end of abx.
Reply: The operation under 4 degrees is to better preserve the composition of the donor flora, so that it does not change as much as possible.
Filtration is used to remove undigested food residues and other substances from faeces. For a number of reasons, many researchers cannot perform FMT immediately after obtaining donor stool. Liquid nitrogen storage is acceptable.
FMT gavage immediately after antibiotic administration may affect its colonization effect and cause certain damage. Studies have shown the effectiveness of this approach (manuscript line 151). It will take at least 14 days for the bacteria to recover [1].
[1] Vicentini, F.A., Keenan, C.M., Wallace, L.E. et al. Intestinal microbiota shapes gut physiology and regulates enteric neurons and glia. Microbiome 9, 210 (2021).

Round 2
Reviewer 3 Report
Authors have submitted a (minor) revised manuscript. My initial assessment has not changed, and I don't think the author's responses to my queries are properly addressed in the revised manuscript. Last paragraph on limitations is not informative.
Author Response
Dear reviewer:
Thank you very much for your suggestions. In fact, we quite agree with what you said plus more information about the way of administration would provide more informative references and make the article more holistic. However, this article is inspired by a previously published article. During the experimental implementation of that article, as beginners, we encountered many difficulties, which made us think that if we had an introductory literature guidance, it would make us more smooth. Therefore, much of this article is based on our own experience. The part you mentioned may be covered later, but it is not within the scope of this article. Thank you very much for your suggestions. I hope we will have more opportunities to learn and communicate in the future. In addition, we have revised the last part according to your suggestion (line273-284):
Conclusion and Perspective
Strategically FMT is the most direct method to change the composition of gut microbiota. Here, we provide a brief overview of FMT protocol and summarized the research progress of FMT. However, This review had some limitations. One limitation is that we only included oral administration, the most common route. The protocol provided in previous part may not be generalizable to other routes, such as rectal FMT. Besides, we have only provided the primary means of FMT and solutions to common problems, and some other difficulties and innovations are not included. These limitations may mean the instructions are only informative for beginners. However, with the rapid progress of gut microbiology, it is hoped that more studies will be conducted in the future to further clarify the application prospects of FMT and seek for more comprehensive and optimized FMT strategies.
